# Nutrient Equivalence of Plant-Based and Cultured Meat: Gaps, Bioavailability, and Health Perspectives

**DOI:** 10.3390/nu17243860

**Published:** 2025-12-10

**Authors:** Jean Demarquoy

**Affiliations:** Université Bourgogne Europe, Institut Agro PAM UMR A 02.102—INRAe, 6 blvd Gabriel, 21000 Dijon, France; jean.demarquoy@ube.fr

**Keywords:** nutrient equivalence, plant-based meat analogs, cultured meat, carnitine, vitamin B12, heme iron, creatine, taurine, bioavailability, fortification, human health

## Abstract

Meat provides high-quality protein and essential micronutrients such as vitamin B12, heme iron, zinc, and selenium, along with conditionally essential compounds including creatine, carnitine, and taurine. Growing concerns over environmental sustainability, animal welfare, and potential health risks associated with excessive meat consumption have spurred the development of plant-based and cultured alternatives intended to replicate the nutritional and sensory attributes of meat. This review critically examines the extent to which these emerging products achieve nutrient equivalence with conventional meat, focusing on essential and conditionally essential nutrients, their bioavailability, and implications for human health. After outlining the physiological importance of nutrients characteristically supplied by meat, the review compares the composition of plant-based meat analogs (PBMAs) and cultured meat prototypes. Differences in fortification strategies, ingredient formulation, and the presence of anti-nutritional factors are discussed in relation to nutrient absorption and utilization. Current PBMAs can approximate protein content but generally provide lower levels and reduced bioavailability of vitamin B12, heme iron, creatine, taurine, and long-chain omega-3 fatty acids unless fortified. Cultured meat offers theoretical potential for compositional optimization through cellular engineering but remains limited by scarce empirical data. Achieving nutrient equivalence with conventional meat thus represents a major scientific, technological, and regulatory challenge. Future progress will depend on integrating nutritional design into product development, validating bioavailability in human studies, and implementing transparent labeling to ensure that next-generation meat alternatives meet both health and sustainability goals.

## 1. Introduction

### 1.1. Meat as a Source of Essential and Conditionally Essential Nutrients

Meat has long played a central role in human diets, supplying both macronutrients and micronutrients in highly bioavailable forms. It is a complete protein source, providing all essential amino acids, and contributes key micronutrients such as vitamin B12, iron, zinc, selenium, and B vitamins in forms readily absorbed by the human body [1,2]. Beyond these, meat contains conditionally essential compounds that are either absent or present in negligible amounts in plant foods, including creatine, carnitine, taurine, and carnosine, all of which support cellular energy metabolism, muscle performance, and antioxidant defenses [3,4].

Globally, meat accounts for a disproportionately high share of nutrient supply compared to its dietary mass. Recent modeling estimated that while meat contributes only 7% of food mass globally, it provides 21% of protein, 29% of dietary fat, and over 50% of vitamin B12 intake [5]. In populations at risk of micronutrient deficiencies, especially vitamin B12, heme iron, and zinc, meat consumption serves as a critical safeguard against malnutrition [6].

Nevertheless, the nutrient composition of meat varies substantially depending on species, feeding practices, fat content, and processing. For instance, grass-fed beef tends to contain higher omega-3 polyunsaturated fatty acids compared to grain-fed beef [7], while processing methods can introduce or increase non-nutritive compounds such as sodium, nitrites, or heterocyclic amines [8]. Furthermore, a robust body of epidemiological evidence consistently associates high consumption of processed and red meats with increased risks of colorectal cancer, cardiovascular diseases, and type 2 diabetes. These associations were formally recognized by the World Health Organization (WHO) and the International Agency for Research on Cancer (IARC), which classified processed meat as carcinogenic to humans (Group 1) and red meat as probably carcinogenic (Group 2A), primarily based on colorectal cancer risk [9]. Mechanistically, these adverse health effects are mediated by several components intrinsic to meat and its processing, including high levels of saturated fatty acids, heme iron, and the formation of secondary metabolites such as *N*-nitroso compounds during curing, smoking, and high-temperature cooking [10]. In parallel, the gut microbial metabolism of dietary choline and carnitine, abundant in red meat, leads to the production of trimethylamine, subsequently oxidized in the liver to trimethylamine *N*-oxide (TMAO), a metabolite implicated in atherosclerosis and cardiometabolic risk. The Food and Agriculture Organization of the United Nations (FAO) and WHO joint expert consultations emphasize moderation in red and processed meat intake within healthy dietary patterns, highlighting their contribution to non-communicable disease burden when consumed in excess. Similarly, the European Food Safety Authority (EFSA) has addressed related risks through evaluations of saturated fat intake, setting population reference intakes to limit cardiovascular risk, as well as through risk assessments of nitrites and nitrates used as food additives, identifying margins of exposure that are of concern for long-term consumer safety [11,12].

### 1.2. Rise of Plant-Based and Cultured Meat Substitutes

Driven by environmental sustainability, animal welfare, and health concerns, alternatives to conventional meat have emerged over the last two decades. Two categories dominate current research and market growth: plant-based analogs and cultured (cultivated) meat.

Plant-based meat analogs are formulated using protein isolates from soy, pea, wheat, or other sources, processed through extrusion, shear-cell, or novel texturization methods to mimic the fibrous structure of meat [13]. While these products may provide comparable protein levels, they often lack or provide only minimal amounts of certain nutrients such as vitamin B12, heme iron, creatine, or taurine. Consequently, fortification strategies (e.g., adding B12, iron, or omega-3 fatty acids) are frequently employed to approach nutritional equivalence [14]. However, the presence of anti-nutrients such as phytates can inhibit mineral absorption, limiting the bioavailability of iron and zinc compared to meat sources [15].

Cultured meat (also called cultivated meat or cell-based meat), derived from ex vivo proliferation of animal muscle progenitor cells on scaffolds, represents a technologically promising but still nascent field. Proof-of-concept products have demonstrated feasibility, but large-scale production remains constrained by cost, bioreactor efficiency, and serum-free media requirements [16,17]. Theoretically, cultured meat could be engineered to match or exceed the nutrient composition of conventional meat, yet current data on its micronutrient profile remain scarce, and challenges persist in replicating complex components such as heme iron and long-chain polyunsaturated fatty acids [18].

Market penetration of these alternatives is increasing: the plant-based meat market grew by 19% globally between 2019 and 2022, while cultured meat has recently gained regulatory approval in Singapore and the United States [19]. However, the regulatory and political landscape remains highly heterogeneous. In December 2023, the government of Italy became the first country in the European Union to enact a law banning the production and sale of cultured (cell-cultured) meat products [20]. The new legislation (Law 172/2023) prohibits not only manufacturing and marketing of cultured meat, but also the use of meat-related terminology (e.g., “steak,” “salami”) to describe plant-based substitutes or cell-based products. This measure reflects a regulatory choice based on the precautionary principle, invoked in the absence of consensus on long-term safety and to preserve traditional livestock and agro-food sectors. Still, issues of cost, consumer acceptance, and labeling remain unresolved [21]. The cost challenge is especially likely to evolve over time: the culture medium currently represents a major share of total production costs for cell-cultured meat. Estimates suggest that 55–95% of the cost of producing cultured meat stems from the price of the culture medium, including growth factors and nutrients. As culture media become more efficient and less expensive, for example, by replacing expensive growth-factor supplements with cells engineered to produce their own growth signals, production costs could drop substantially [22]. On the consumer side, acceptance remains limited: many individuals perceive cultured meat as “unnatural,” and willingness to pay remains contingent on price parity with conventional meat. Labeling concerns also persist, as regulators, producers and consumers debate how to classify and name these novel products [23]. Importantly, without careful nutritional design, the replacement of meat with alternatives may lead to nutrient deficiencies, especially for populations dependent on meat as a major source of bioavailable micronutrients.

### 1.3. Objectives and Scope of the Review

Although a rapidly expanding body of literature has addressed the sustainability metrics, environmental footprint, and consumer acceptance of meat alternatives, comparatively few reviews have systematically examined the critical issue of nutrient equivalence relative to conventional meat. Yet, nutritional adequacy is a key determinant of both long-term health outcomes and consumer trust in these emerging food products. Against this background, the present review aims to critically evaluate how cultured and plant-based meat substitutes compare with traditional meat in terms of nutrient composition, bioavailability, and potential health implications.

This review was conducted as a narrative and critical synthesis of the available literature. The bibliographic search was performed using multiple complementary databases to ensure broad coverage of both nutritional and biotechnological aspects of meat alternatives. Scientific publications were primarily retrieved from PubMed/MEDLINE, Web of Science, Scopus, and Google Scholar. Regulatory and safety-related documents were accessed from EFSA, WHO, and FAO official databases. The search strategy combined keywords related to plant-based meat, cultured meat, nutrient composition, bioavailability, fortification, health outcomes, and sustainability. Reference lists of key articles and recent reviews were also screened manually to identify additional relevant publications.

Specifically, this review first revisits the physiological role of essential and conditionally essential nutrients supplied by meat, then assesses the nutrient composition of plant-based and cultured meat substitutes, with particular attention to fortification strategies and ingredient formulation. It further examines the available evidence on nutrient bioavailability and metabolic utilization across these food matrices, evaluates the potential nutritional benefits and risks associated with the partial or complete replacement of conventional meat with alternatives, and finally identifies key research gaps and future perspectives for the development of nutritionally optimized, consumer-accepted, and environmentally sustainable meat substitutes.

By integrating nutritional science with public health considerations, this review seeks to provide a framework for evaluating whether meat alternatives can realistically achieve equivalence with conventional meat in supporting human health.

## 2. Nutrients Provided by Meat and Their Physiological Importance

The composition of plant-based and cultured meat products is not intrinsically fixed but remains highly dependent on formulation choices, processing conditions, and fortification strategies. Their nutritional profiles can therefore vary substantially between products, brands, and production batches. As a result, direct comparisons with raw or minimally processed meat should be interpreted with caution and should not be not overstated, as they often reflect specific technological designs rather than inherent nutritional equivalence.

### 2.1. Conditionally Essential Nutrients: Carnitine, Creatine, Taurine

Meat provides several bioactive nutrients that are not strictly essential in the diet, but whose limited endogenous synthesis may not meet physiological requirements under certain conditions, particularly in infants, during pregnancy, or in individuals with specific metabolic demands.

**Carnitine** (L-carnitine) plays a central role in mitochondrial fatty acid oxidation by shuttling long-chain acyl-CoAs across the inner mitochondrial membrane via the carnitine shuttle. Omnivorous diets provide between 20 and 200 mg/day, while strict vegetarian diets supply <5 mg/day, relying mainly on endogenous synthesis from lysine and methionine [4]. Plasma carnitine concentrations are consistently lower in vegetarians than in omnivores, though usually within the normal physiological range [24]. Supplementation has shown benefits in metabolic disorders, cardiovascular disease, and exercise recovery, highlighting its conditional essentiality [25].

**Creatine**, synthesized from arginine, glycine, and methionine, is critical for cellular energy buffering through the creatine-phosphocreatine system. Approximately half of the daily requirement (~2 g/day) is supplied by the diet, almost exclusively from meat and fish [26]. Vegetarians and vegans exhibit lower muscle creatine stores, and these reduced stores may impair short-term high-intensity exercise performance; supplementation restores levels and improves function [27]. Beyond muscle, creatine supports neural development and cognitive function, with potential clinical applications in neurodegenerative diseases [28].

**Taurine** (2-aminoethanesulfonic acid) is abundant in animal tissues and virtually absent in plants. It contributes to bile salt conjugation, osmoregulation, calcium signaling, and antioxidative defense [29]. While humans synthesize taurine from cysteine, endogenous production may be insufficient in preterm infants or under stress conditions. Omnivorous diets provide ~40–400 mg/day, whereas vegan diets provide negligible amounts [30]. Low taurine intake has been implicated in altered lipid metabolism and retinal function, underscoring its importance despite its non-essential status.

### 2.2. Key Vitamins: Vitamin B12 and Vitamin D

**Vitamin B12 (cobalamin)** is exclusively produced by microorganisms and accumulates in animal tissues. Meat, fish, and dairy are primary dietary sources, whereas unfortified plant-based foods contain negligible amounts [31]. B12 is required for methionine synthase and methylmalonyl-CoA mutase, enzymes essential for one-carbon metabolism and fatty acid oxidation. Deficiency causes megaloblastic anemia, neuropathy, and elevated homocysteine, with increased risk in vegetarians and vegans [32]. Average B12 concentrations in beef muscle are ~0.7–1.5 µg/100 g [33].

**Vitamin D** is present in small amounts in meat, especially fatty cuts, though its contribution is modest compared to oily fish and fortified foods. Nonetheless, beef and pork provide measurable 25-hydroxyvitamin D, the metabolically active form, which has greater bioactivity than vitamin D3 [34]. This metabolite may contribute to vitamin D status in omnivorous diets, though plant-based diets lack this source.

### 2.3. Minerals and Trace Elements: Heme Iron, Zinc, Selenium

**Iron** in meat occurs largely as heme iron, which accounts for 40–60% of total iron in red meat. Heme iron is absorbed at rates of 15–35%, compared with 2–20% for non-heme iron from plants. This difference explains why meat consumption strongly protects against iron deficiency anemia, particularly in women and children [35]. Excessive heme iron intake, however, has been linked with oxidative stress and colorectal cancer risk [11].

**Zinc** is abundant in meat, with absorption rates of 25–40%, significantly higher than in plant foods due to the absence of phytate complexes [15]. Zinc is essential for immune function, growth, and reproduction; vegetarians often exhibit lower zinc status compared to omnivores [36].

**Selenium** is supplied by meat in the form of selenomethionine and selenocysteine, both highly bioavailable. Selenium is incorporated into selenoproteins critical for antioxidant defense and thyroid function. Meat, poultry, and fish are important contributors to selenium intake in regions where soil selenium is low [37].

### 2.4. Long-Chain Polyunsaturated Fatty Acids: DHA and EPA

While alpha-linolenic acid (ALA) from plant sources can be endogenously converted into eicosapentaenoic acid (EPA) and docosahexaenoic acid (DHA), conversion efficiency is low (<10% for EPA and <1% for DHA) [38]. Meat, especially from grass-fed ruminants and game, provides modest but significant amounts of preformed EPA and DHA [7]. These fatty acids support cardiovascular, cognitive, and retinal health, making their presence in meat relevant in diets lacking marine products.

### 2.5. Other Relevant Bioactive Compounds

Additional compounds in meat may contribute to human health. Carnosine, a dipeptide of beta-alanine and histidine, is abundant in skeletal muscle and acts as an intracellular pH buffer and antioxidant [39]. **Anserine**, structurally related to carnosine, provides similar functions but is less abundant in human muscle. These compounds are absent in plants and have been linked to exercise performance and neuroprotection [40]. Table 1 summarizes the comparative nutrient composition of meat, PBMAs, and cultured meat.

## 3. Nutrient Composition of Meat Alternatives

### 3.1. Plant-Based Meat Analogs: Soy, Pea, Wheat and Emerging Sources

Plant-based meat analogs (PBMAs) are designed to replicate the sensory and nutritional attributes of traditional meat using plant-derived proteins, fats, and additives. Common protein bases include soy, pea, wheat gluten (seitan), and mycoprotein, processed through extrusion or shear-cell technologies to generate fibrous textures [46].

From a macronutrient perspective, most PBMAs are engineered to approximate the protein content of meat (15–25 g/100 g). However, protein quality differs: although soy and pea proteins are relatively high in indispensable amino acids. Methionine is the limiting amino acid in legumes, and lysine is limiting in cereals [41]. The digestible indispensable amino acid score (DIAAS) of soy protein isolate is ~0.9, compared with >1.0 for beef, indicating lower bioavailability [42].

Micronutrient composition diverges more substantially. PBMAs contain little to no vitamin B12, heme iron, carnitine, creatine, taurine, or long-chain *n*-3 polyunsaturated fatty acids unless fortified. Iron and zinc may be added in non-heme forms, but their absorption is reduced by phytates and polyphenols present in plant matrices [43]. Similarly, plant-based omega-3 fortification typically involves alpha-linolenic acid (ALA) or algal-derived DHA, which may have different bioavailability compared to meat sources [38].

Recent surveys indicate that although PBMAs are increasingly fortified with vitamin B12, iron, zinc, and DHA, fortification strategies vary widely among brands and markets, and regulatory frameworks differ between regions [14,47]. Thus, nutrient equivalence across PBMAs is inconsistent, and consumers may not achieve the same micronutrient intake as they would when consuming traditional meat.

### 3.2. Cultured Meat: Current Knowledge, Variability, and Technological Challenges

Cultured meat is produced by proliferating animal myoblasts or stem cells in bioreactors, followed by differentiation on scaffolds to generate muscle-like tissue [16]. Early prototypes have demonstrated technical feasibility, yet comprehensive and systematic data on their nutrient composition remain limited.

Theoretically, cultured meat can replicate muscle protein and amino acid composition of traditional meat, as cells originate from animal tissues. However, micronutrient content is less predictable. Unlike animals that acquire nutrients through diet and metabolism, cultured cells depend entirely on the growth medium. For instance, vitamin B12, creatine, carnitine, and long-chain *n*-3 fatty acids are not synthesized by mammalian cells in vitro and must be supplied exogenously [18]. Whether such nutrients can be stably incorporated into cultured muscle tissue at physiologically relevant levels remains under investigation [46].

However, most cultured meat products under development consist primarily of muscle cells and generally lack adipose tissue. In conventional meat, fat plays a crucial role in determining flavor, juiciness, and overall sensory quality, as well as contributing key nutrients such as essential fatty acids and fat-soluble vitamins. The absence of adipose tissue in cultured meat therefore presents both nutritional and sensory challenges. To address this limitation, some research groups are exploring co-culture strategies with adipocytes or the incorporation of cultured fat to more closely mimic the composition and eating experience of conventional meat [48].

Fatty acid composition is another challenge. Meat lipid profiles reflect animal feeding practices (grass- vs. grain-fed), but in cultured systems, lipid provision depends on medium supplementation or co-culture with adipocytes [49]. Engineering cultured meat to contain adequate levels of EPA and DHA is of particular interest but remains unresolved. Similarly, heme iron, which contributes to iron nutrition and flavor chemistry in conventional meat, may be absent unless supplemented [17]. Beyond the production of cultured muscle tissue, increasing attention is now being directed toward the development of cultured fat, which plays a central role in the sensory quality, nutritional profile, and consumer acceptance of meat products. Adipose tissue is a major determinant of flavor, juiciness, and texture, and its absence has been identified as a key limitation of early cultured meat prototypes composed predominantly of myocytes. Recent advances in adipocyte differentiation, three-dimensional co-culture systems, and edible scaffolding have enabled the production of structured cultured fat tissues and muscle–fat composites that more closely resemble conventional meat [50]. From a nutritional perspective, cultured fat offers the theoretical possibility of tailoring fatty-acid profiles through controlled lipid supply and metabolic programming, potentially enabling reductions in saturated fatty acids and enrichment with monounsaturated or long-chain omega-3 polyunsaturated fatty acids [51].

Overall, cultured meat has potential to be nutritionally tailored, but empirical data are currently limited to prototypes, and compositional equivalence to conventional meat has not yet been established.

### 3.3. Fortification, Supplementation, and Formulation Strategies

Both plant-based and cultured meat alternatives rely on fortification strategies to address nutritional gaps. Common fortifications include:

**Vitamin B12**: vitamin B12 is naturally absent from plant matrices and not synthesized by mammalian cells used in cultured meat systems, as this vitamin is produced exclusively by specific microorganisms. Consequently, vitamin B12 in both plant-based and cultivated meat products must be supplied through synthetic fortification or via ingredients derived from microbial fermentation. However, vitamin B12 is sensitive to heat and oxidative processing, raising concerns about its stability during extrusion, storage, and cooking. [31].

**Iron**: Iron is another key target of fortification in plant-based and cultured meat alternatives and is typically added in the form of non-heme iron salts such as ferric pyrophosphate, ferrous fumarate, or ferrous sulfate. These forms are chemically stable and cost-effective but generally exhibit lower intestinal absorption than heme iron naturally present in animal meat. To address both nutritional and sensory limitations, heme-mimicking ingredients produced through precision fermentation, such as soy leghemoglobin used in products like the Impossible™ Burger, have been developed to enhance iron bioavailability while also contributing to meat-like color and flavor [52].

**Zinc and selenium**: Zinc and selenium are also commonly included in fortification strategies for plant-based and cultured meat alternatives, typically added in inorganic forms (e.g., zinc sulfate, sodium selenite) or as organic complexes such as zinc gluconate or selenomethionine [53,54]. While these fortificants can effectively increase total micronutrient content, their intestinal absorption is often lower and more variable than that of zinc and selenium naturally present in animal tissues. In plant-based matrices, bioavailability is further constrained by the presence of phytates, dietary fiber, and polyphenols, which chelate divalent minerals and reduce their uptake.

**Omega-3 fatty acids**: Long-chain omega-3 polyunsaturated fatty acids, particularly eicosapentaenoic acid (EPA) and docosahexaenoic acid (DHA), are largely absent from most plant-based formulations and are therefore commonly supplied through fortification using algal-derived oils. This strategy allows partial replication of the fatty acid profile of meat and fish without reliance on marine sources and is compatible with vegan formulations. However, EPA and DHA are highly susceptible to oxidative degradation during high-temperature processing, storage, and cooking, which may substantially reduce their effective content at consumption [55]. In addition, interactions with the plant-based matrix and the presence of pro-oxidant compounds can further compromise lipid stability [47].

Emerging approaches involve bioengineering and fermentation to enhance nutritional value. Precision fermentation is being explored to produce heme proteins, B12, or other bioactives directly in the product [56]. Similarly, co-culturing adipocytes with myocytes in cultured meat could improve lipid composition [46].

Despite these strategies, challenges remain in ensuring nutrient stability during processing, storage, and cooking. Moreover, fortification does not always guarantee equivalent **bioavailability (see below)**.

## 4. Nutrient Equivalence and Bioavailability

### 4.1. Comparative Nutrient Profiles Across Traditional, Plant-Based, and Cultured Meats

When comparing nutrient composition, traditional meat is a unique source of both essential and conditionally essential compounds. Plant-based meat analogs (PBMAs) can approximate protein content but generally lack vitamin B12, creatine, carnitine, taurine, and heme iron unless fortified [47]. Cultured meat has the theoretical potential to replicate the protein and amino acid profile of conventional meat but remains poorly characterized with respect to micronutrient content [18,46].

Comparisons across categories reveal that while protein quantity may be matched, micronutrient density diverges significantly. For example, vitamin B12 content in beef averages 1–2 µg/100 g, meeting ~40% of daily requirements, whereas most PBMAs contain none unless fortified [31]. Similarly, heme iron in red meat provides 40–60% of total iron and is highly bioavailable, while PBMAs contain only non-heme iron, often with reduced absorption [43]. Cultured meat prototypes have not yet demonstrated consistent inclusion of B12 or heme iron [49].

### 4.2. Influence of the Food Matrix and Processing on Nutrient Retention

Nutrient bioavailability is not determined by nutrient content alone but by interactions within the food matrix. In meat, the presence of muscle proteins enhances the absorption of non-heme iron from other foods (the “meat factor”), an effect attributed to peptides generated during digestion that promote iron solubility and uptake [57]. Such synergistic effects are absent in PBMAs, and in fact, anti-nutritional compounds in plants may reduce mineral bioavailability [15].

Processing also affects nutrient retention. High-temperature cooking of meat can reduce taurine and creatine content by up to 50%, but these compounds are largely absent in plant-based alternatives regardless of cooking [45]. In cultured meat, nutrient stability during processing and cooking has yet to be evaluated but represents a critical research gap [16].

### 4.3. Anti-Nutritional Factors and Their Impact on Absorption

Plant-derived proteins often come with anti-nutritional factors, including phytates, tannins, and oxalates, which bind minerals and reduce their absorption. For instance, phytate can reduce zinc absorption by up to 50% in high-phytate diets [36]. This explains why vegetarians often exhibit lower serum zinc despite adequate intake. Similarly, iron absorption from legumes may be as low as 5%, compared with 15–35% from heme iron in meat [43]. While fortification can restore nutrient levels in PBMAs, absorption remains limited by the matrix.

Cultured meat, depending on its growth media and scaffolding, may not contain these inhibitors. However, whether cultured meat will replicate the “meat factor” that enhances mineral absorption is unknown [46].

## 5. Health Implications of Meat Replacement

### 5.1. Risks of Nutrient Deficiencies in Plant-Based Diets

Replacing meat with plant-based meat analogs (PBMAs) or cultured meat carries the risk of nutrient inadequacies if products are not nutritionally optimized. Evidence shows that vegetarians and especially vegans are at increased risk of vitamin B12 deficiency, as this vitamin is virtually absent from plant sources without fortification [32]. Deficiency can result in megaloblastic anemia, neuropathy, and cognitive impairment [31].

Iron deficiency is consistently reported as more prevalent among populations consuming diets that exclude meat, largely due to the absence of highly bioavailable heme iron and the lower intestinal absorption of non-heme iron from plant sources [35]. Unlike heme iron, whose absorption is relatively efficient and less affected by dietary factors, non-heme iron absorption is strongly influenced by the food matrix and is frequently inhibited by phytates, polyphenols, and dietary fibers that are abundant in plant-based diets. As a consequence, total iron intake may appear adequate on a quantitative basis, while functional iron status remains insufficient [58]. At the population level, this reduced bioavailability translates into a higher risk of depleted iron stores and iron deficiency anemia in individuals adhering to vegetarian or vegan dietary patterns [59]. Women of reproductive age, children, and adolescents are particularly vulnerable due to their elevated iron requirements associated with growth, menstruation, and increased erythropoiesis. In these groups, insufficient iron intake or poor absorption is associated with impaired cognitive development, reduced physical performance, altered immune function, and, in the case of pregnancy, increased risks of preterm birth and low birth weight [60]. These considerations highlight that the replacement of conventional meat with alternative products must be carefully managed to ensure not only adequate total iron content, but also sufficient bioavailability at the level required to meet physiological demands across the life course.

Similarly, the absence of creatine, carnitine, and taurine in unfortified plant-based diets may influence muscle metabolism, neurodevelopment, and cardiovascular function [28,61]. Although the body can synthesize these compounds, studies in vegetarians consistently show lower tissue levels, with potential implications for exercise performance and metabolic resilience [27,30].

Finally, zinc and selenium intakes are often lower in vegetarian and vegan diets because of reduced bioavailability from plant sources, potentially compromising immune function and thyroid metabolism [36,37].

### 5.2. Supplementation and Fortification Strategies

Nutrient deficiencies can be mitigated through supplementation or fortification. PBMAs are increasingly fortified with vitamin B12, iron, zinc, and omega-3 fatty acids [62]. For example, soy leghemoglobin has been used to provide iron with meat-like sensory properties in the Impossible™ Burger [63]. Fortification with algal-derived DHA and EPA can help offset the absence of long-chain *n*-3 fatty acids [38].

Cultured meat offers opportunities for nutrient tailoring, such as incorporating omega-3 fatty acids via culture media or co-culture with adipocytes, and adding vitamin B12 or creatine directly to growth substrates [46]. However, whether such approaches will achieve physiological equivalence in terms of bioavailability remains uncertain.

### 5.3. Potential Advantages of Meat Substitutes

Beyond risks, meat substitutes may confer health advantages. PBMAs typically contain less saturated fat and cholesterol than red meat, which may reduce cardiovascular risk [64]. They also provide more dietary fiber, absent in meat, which supports gut health and reduces risk of colorectal cancer [44].

Another debated aspect is trimethylamine-*N*-oxide (TMAO) production. Meat, particularly red meat, provides precursors such as carnitine and choline that can be metabolized by gut microbiota to TMAO, a compound associated with increased cardiovascular risk in some studies [65]. PBMAs do not contribute these precursors to the same extent.

Cultured meat could also be designed to have favorable lipid profiles (e.g., higher polyunsaturated fatty acids, lower saturated fats), potentially improving cardiovascular outcomes compared with conventional red meat [16].

### 5.4. Long-Term Health Outcomes and Research Gaps

While short-term nutrient comparisons are increasingly available, long-term health outcomes of diets rich in PBMAs or cultured meat remain poorly studied. Epidemiological data on vegetarians suggest that well-planned meat-free diets can be compatible with health, but often require supplementation, and risks of deficiencies persist if not properly addressed [66].

For cultured meat, no long-term consumption studies exist, and its nutritional equivalence remains hypothetical. Research is needed to determine whether fortification strategies in PBMAs and cultured meat truly achieve equivalence in bioavailability and whether such products influence chronic disease risk differently than conventional meat. Key gaps include the lack of longitudinal studies on nutrient status in populations regularly consuming PBMAs or cultured meat, as well as the absence of clinical trials that compare health outcomes between fortified and unfortified alternatives. Further investigations are also required to clarify the metabolic and safety effects of novel ingredients such as leghemoglobin or algal oils. In addition, more research should examine how meat alternatives interact with the gut microbiota, with particular attention to iron and carnitine metabolism.

## 6. Perspectives and Future Directions

### 6.1. Designing Nutritionally Optimized Meat Alternatives

The future of meat alternatives will depend not only on replicating the sensory attributes of meat but also on achieving nutritional equivalence. To date, most plant-based meat analogs (PBMAs) are formulated to match protein quantity but remain deficient in vitamin B12, heme iron, creatine, taurine, and long-chain omega-3 fatty acids unless fortified [47]. Next-generation PBMAs are likely to combine protein blends (soy, pea, wheat, mycoprotein, insect, or algal proteins) with precision fermentation ingredients such as heme proteins, B12, or DHA to improve both sensory and nutritional value [67].

Cultured meat offers unique opportunities for nutrient tailoring. By modifying growth media or co-culturing muscle cells with adipocytes, it may be possible to enhance levels of omega-3 fatty acids, optimize fatty acid ratios, and supplement micronutrients otherwise absent from in vitro tissues [68]. However, this approach requires demonstration of stable nutrient incorporation and proof of equivalent bioavailability in vivo, which has yet to be validated.

### 6.2. Regulatory Frameworks, Labeling, and Consumer Communication

As meat alternatives become more widespread, regulatory definitions and labeling standards will be crucial to ensure transparency and protect consumers. In the European Union, the use of terms such as “burger” or “sausage” for plant-based products has been subject to debate, with proposals to restrict their use to animal-derived foods [69]. In the United States, cultured meat products fall under joint regulation by the USDA and FDA, which oversee labeling and safety assessments [70]. Singapore has already authorized the sale of cultured chicken, establishing the first precedent for market approval [71].

Nutrient labeling is particularly important for PBMAs, since their composition varies widely. Studies show that consumers often assume equivalence with conventional meat, even when key micronutrients are lacking [72]. Mandatory labeling of fortification status (e.g., B12, iron, DHA) may help consumers make informed choices and mitigate deficiency risks.

### 6.3. Consumer Acceptance and Cultural Considerations

Consumer attitudes remain a major determinant of adoption. Surveys indicate that acceptance of plant-based meat is relatively high, particularly among flexitarians and younger demographics. Acceptance of cultured meat is more variable, often influenced by perceptions of “unnaturalness,” food safety concerns, and cultural norms [73].

Long-term adoption will require building trust in safety and nutritional adequacy. Clear communication on health benefits (e.g., lower saturated fat, fiber enrichment), sustainability gains, and nutrient fortification will be critical. In addition, ensuring affordability and accessibility will be key for public health impact beyond niche markets.

### 6.4. Research Priorities for Nutrient Equivalence and Health Outcomes

Despite rapid technological progress, several critical research gaps must be addressed before plant-based and cultured meat alternatives can be considered nutritionally equivalent to conventional meat in a robust, evidence-based manner. Comprehensive and standardized nutrient profiling of cultured meat prototypes remains insufficient, particularly with regard to micronutrients, trace elements, lipid classes, bioactive compounds, and processing-induced modifications. Many currently available data are still derived from conceptual formulations or laboratory-scale systems rather than market-ready products.

Well-controlled human bioavailability studies are urgently needed to directly compare fortified plant-based meat analogs, cultured meat, and conventional meat under standardized dietary conditions. Such trials should specifically assess the absorption and metabolic utilization of key nutrients of concern, including vitamin B12, iron, zinc, selenium, and long-chain omega-3 fatty acids, while accounting for food matrix effects and interindividual variability.

Long-term intervention and observational studies are also required to evaluate the real-world health consequences of substituting conventional meat with alternative products. These should include longitudinal assessments of micronutrient status, cardiometabolic risk factors, inflammatory markers, and gut microbiota composition and function, given the central role of the microbiota in metabolic and immune regulation.

The influence of food matrix structure and processing must be examined more systematically, particularly with respect to whether novel formulations can reproduce the so-called “meat factor” known to enhance non-heme iron absorption in mixed meals. How protein organization, lipid structuring, and processing-induced chemical changes influence mineral bioaccessibility remains a major unresolved challenge.

Consumer-focused intervention studies are essential to determine whether fortification, reformulation, and labeling strategies translate into measurable improvements in nutritional status under real-world consumption conditions. These studies should integrate behavioral, socioeconomic, and dietary adherence dimensions alongside biological endpoints.

Addressing these priorities will require close interdisciplinary collaboration among nutrition scientists, food technologists, microbiologists, clinicians, and regulatory bodies, and will be central to establishing scientifically sound frameworks for the safe, effective, and nutritionally adequate integration of meat alternatives into human diets.

## 7. Conclusions

Traditional meat remains a uniquely dense source of essential and conditionally essential nutrients, including vitamin B12, heme iron, zinc, selenium, creatine, carnitine, and taurine, many of which are absent or present in negligible amounts in unfortified plant foods. While plant-based meat analogs (PBMAs) can match protein content and often provide fiber and lower levels of saturated fat, they generally fall short in supplying bioavailable micronutrients unless fortified. Cultured meat holds the theoretical potential to replicate or even enhance the nutrient profile of conventional meat, but empirical data remain scarce, and key issues of micronutrient incorporation and bioavailability are unresolved.

The evidence reviewed highlights that nutrient equivalence cannot be assumed simply from compositional claims. Differences in bioavailability, the influence of the food matrix, and the absence of “meat factor” effects in PBMAs all contribute to divergent nutritional outcomes. Risks of deficiencies in vitamin B12, iron, zinc, and long-chain *n*-3 fatty acids remain significant in diets relying heavily on inadequately fortified alternatives. At the same time, meat substitutes may confer advantages by providing fiber, reducing intake of saturated fat, and limiting exposure to compounds associated with chronic disease risk such as TMAO precursors and nitrosating agents.

Future innovation in meat alternatives should therefore focus on nutritional optimization in parallel with sensory and sustainability goals. This includes systematic fortification strategies, precision fermentation of critical micronutrients, and engineering of cultured meat with tailored lipid and micronutrient profiles. Transparent labeling and harmonized regulatory standards will be essential to guide consumers, while robust clinical trials are needed to determine whether nutrient equivalence translates into equivalent health outcomes. Ultimately, the transition toward alternative proteins presents both opportunities and challenges: ensuring that nutritional adequacy is maintained will be pivotal to their role in supporting long-term human health.

## Figures and Tables

**Table 1 nutrients-17-03860-t001:** Comparative Nutrient Composition of Meat, PBMAs, and Cultured Meat.

Nutrient	Conventional Meat	Plant-Based Meat Analogs (PBMAs)	Cultured Meat (Prototype Data)	Key References
Protein (g/100 g)	18–25	15–25	18–22	[41,42]
Protein Quality (DIAAS)	>1.0	0.7–0.9 (soy/pea isolates)	Expected > 1.0	[36,37,41,42]
Vitamin B12 (µg/100 g)	0.7–1.5	0 (unless fortified)	0 (must be supplemented in medium)	[31]
Iron (mg/100 g)	2–3 (red meat)	2–4	2–3 (uncertain form)	[35,43]
Iron Form	40–60% heme	Non-heme only	Likely non-heme unless engineered	[35]
Zinc (mg/100 g)	4–6	2–4 (lower absorption)	4–6 (depends on medium)	[15,36]
Carnitine (mg/100 g)	20–200	0	0 (must be supplemented)	[24,25]
Creatine (g/100 g)	~0.4–0.5	0	0 (must be supplemented)	[26,27]
Taurine (mg/100 g)	40–400	0	0 (must be supplemented)	[29]
EPA + DHA (mg/100 g)	20–80 (higher in grass-fed)	0 (unless algal oil fortified)	0 (unless engineered)	[7,38]
Fiber (g/100 g)	0	2–6	0	[44]
Saturated Fat (g/100 g)	5–9 (beef)	1–4	Variable, usually lower than beef	[45]

DIAAS: Digestible Indispensable Amino Acid Score. For a more detailed and standardized assessment of food composition, reference may be made to national food composition databases, including those of the USDA (https://fdc.nal.usda.gov) and ANSES (https://ciqual.anses.fr/) for instance.

## Data Availability

No new data were created or analyzed in this study. Data sharing is not applicable to this article.

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
