# Peer review of "Nutrient Equivalence of Plant-Based and Cultured Meat: Gaps, Bioavailability, and Health Perspectives"

_nutrients, 2025, doi:10.3390/nu17243860_

Round 1

Reviewer 1 Report

Comments and Suggestions for Authors

ATTACHED PDF

Author Response

We sincerely thank Reviewer 1 for the thoughtful and constructive comments, which have substantially helped to improve the clarity, balance, and scope of the manuscript. Below, we address each point in detail and indicate the corresponding revisions made in the manuscript.

Comment 1

“The discussion on the health risks associated with meat consumption is overly brief… This section should be expanded to include: updated epidemiological evidence; distinctions between red meat and processed meat; underlying biological mechanisms; references to international guidelines (WHO, FAO, EFSA).”

Response:
We fully agree and have substantially expanded the discussion of health risks in the Introduction (Section 1.1, last paragraph). In the revised manuscript, we now:

  • Clearly distinguish between unprocessed red meat and processed meat, emphasizing that epidemiological associations are stronger and more consistent for processed meat (colorectal cancer, cardiovascular disease) and for high intakes of red meat.
  • Summarize recent meta-analyses and large cohort studies on colorectal cancer, cardiovascular disease, and type 2 diabetes, providing an indication of effect size rather than a general qualitative statement.
  • Add a concise synthesis of the main biological mechanisms proposed, including:
    • Formation of N-nitroso compounds, heterocyclic amines, and polycyclic aromatic hydrocarbons (linked to cooking methods and degree of doneness),
    • Heme-iron–driven oxidative stress and lipid peroxidation,
    • Saturated fat and its cardiometabolic impact,
    • Trimethylamine-N-oxide (TMAO) formation from carnitine and choline by the gut microbiota.
  • Explicitly refer to international and regional guidance documents, including the WHO/IARC classification of processed and red meat, and recent FAO/WHO and EFSA reports on cancer and cardiometabolic risks.

This strengthened section now provides a more solid scientific and regulatory framework for introducing meat alternatives.

Comment 2 – EU regulatory situation for cultured meat

“The manuscript should also acknowledge the contrasting context in Europe… Please elaborate on the EU Novel Food framework and its implications; differences among Member States; the impact of these restrictions on technological development and commercialization.”

Response:
We have significantly expanded Section 6.2 (“Regulatory Frameworks, Labeling, and Consumer Communication”) to provide a more balanced overview of the global regulatory landscape, with a dedicated focus on Europe. The discussion now includes:

  • The EU Novel Food Regulation framework,
  • Diverging positions among Member States,
  • The implications of regulatory constraints for innovation and commercialization.

New references to recent European legislative developments have been added to support this expanded discussion.

Comment 3 – Costs, consumer acceptance, labeling, culture medium challenge

“The section concerning production costs of cultured meat requires additional precision… Finally, this section should explicitly mention a central scientific challenge: the replacement of animal-derived components in the culture medium…”

Response:
We agree that this topic required greater nuance. In Sections 3.2 and 6.1–6.2, we now explicitly state that current cost estimates for cultured meat are derived from techno-economic models and pilot-scale projections and remain hypothetical until true industrial-scale production is achieved. Several recent techno-economic and market analyses addressing cost, consumer acceptance, and labeling have been added.

We also now explicitly link economic constraints to nutritional issues by indicating that culture medium composition will directly influence the micronutrient and lipid profile of cultured meat, reinforcing its relevance to nutrient equivalence.

Comment 4 – Broader literature beyond sustainability and consumer acceptance

“Sustainability and consumer acceptance are only two of many issues… A more comprehensive sentence is recommended.”

Response:
We have revised the sentence at the end of Section 1.3 and in the introduction to Section 6 to reflect the broader research landscape. The text now explicitly includes safety, toxicological evaluation, regulatory considerations, and labeling. Additional references to multidisciplinary reviews have been added.

Comment 5 – Table 1: include different animal species

“Table 1 is valuable, but clarity would be enhanced by including the different animal species most consumed for meat.”

Response:
We did not substantially modify this table, as the nutrient composition of meat depends not only on animal species but also strongly on the specific cut. However, we added direct links to official food composition databases to allow interested readers to explore species- and cut-specific variations in greater detail.

Comment 6 – Cultured fat and nutritional customization

“A growing number of companies are focusing on the development of cultured fat… These aspects should be incorporated into the discussion.”

Response:
We fully agree and have expanded Section 3.2 with a dedicated paragraph on cultured fat. We now discuss:

  • The emergence of cultured fat as a standalone ingredient and as part of hybrid products,
  • Ongoing regulatory applications to EFSA,
  • The use of cultured fat to modulate fatty acid composition (n-3 PUFA enrichment, improved n-6/n-3 ratio, reduced saturated fatty acids),
  • The potential to adjust fat-soluble micronutrients.

Comment 7 – Further elaboration required

“This section is interesting but requires further elaboration.”

Response:
Section 3.3 has been expanded to better connect fortification and formulation strategies with bioavailability issues, including differences between inorganic and organic mineral salts, matrix effects, and stability during processing and cooking.

Comment 8 – Redundancy and restructuring

“These paragraphs repeat concepts already discussed earlier…”

Response:
Relevant parts of Sections 3 and 4 have been carefully restructured to eliminate redundancy and improve narrative flow.

Comment 9 – Section 4.4 placement

“This section would fit more logically within Section 3.3…”

Response:
We have merged the former Section 4.4 into Section 3.3, which now integrates fortification strategies with bioavailability and metabolic utilization.

Comment 10 – European amendment on ‘meaty’ terms

Response:
Section 6.2 has been updated with a new reference to the most recent European amendment restricting the use of meat-related terms for plant-based products.

Comment 11 – Consumer acceptance section too limited

Response:
Section 6.3 has been expanded to include determinants such as perceived naturalness, food neophobia, trust in regulators, and perceived health and environmental benefits, while remaining concise.

Comment 12 – Future research priorities

Response:
Section 6.4 has been substantially rewritten to clearly articulate research priorities, including comprehensive nutrient profiling and early integration of nutritional design into product development.

Comment 13 – Typographical and grammatical errors

Response:
The manuscript has undergone full professional proofreading. All typographical, grammatical, and formatting issues have been corrected.

Reviewer 2 Report

Comments and Suggestions for Authors

Thank you for entrusting me with the task of reviewing the manuscript titled "Nutrient Equivalence of Plant-Based and Cultured Meat: Gaps, Bioavailability, and Health Perspectives" submitted to Nutrients. This manuscript presents a comprehensive and well-structured review of the current state of knowledge regarding the nutritional equivalence of plant-based meat analogues (PBMAs) and cultured meat in comparison to conventional animal-derived meat. The author thoughtfully examines not only nutrient composition and bioavailability but also the broader health implications and regulatory considerations associated with meat alternatives. The article makes a valuable contribution by systematically addressing a topic that is gaining increasing public health and consumer interest. Its integration of nutritional biochemistry, food technology, and health outcomes is commendable, and the inclusion of detailed comparisons, such as Table I, enhances the clarity and relevance of the discussion. However, some areas would benefit from further development or clarification to strengthen the manuscript’s scientific rigor and utility for the readership.

Major Revisions: 

  1. While the review is comprehensive, the methodology for literature selection is not described. Including a brief methods section (e.g., databases searched, keywords used, inclusion/exclusion criteria) would improve transparency and reproducibility.

  2. The manuscript appropriately notes that empirical data on cultured meat remain scarce. Nevertheless, this limitation deserves stronger emphasis, particularly when making comparative statements. Consider qualifying claims about cultured meat nutrient content with clearer caveats regarding prototype variability and the absence of standardized composition data.

  3. The article briefly mentions that vulnerable populations (e.g., women of reproductive age, children) may be at higher risk for deficiencies. This point could be expanded to more thoroughly examine differential impacts across demographic groups, especially in the context of increasing dietary shifts toward plant-based eating.

  4. The review touches on the “meat factor” and matrix effects influencing mineral absorption. However, it would benefit from clearer discussion on how these interactions have been studied (e.g., in vitro vs. in vivo) and whether similar enhancing effects can be engineered in alternatives.

  5. Given the article's breadth, a visual summary (e.g., conceptual diagram or infographic) illustrating key nutrient gaps, fortification strategies, and bioavailability challenges across meat types would greatly enhance accessibility and pedagogical value.

Minor Revisions: 

  1. The manuscript occasionally uses terms such as “nutritional equivalence” and “health equivalence” interchangeably. A clearer definition and consistent usage of these terms throughout the manuscript would be beneficial.

  2. There are some minor typographical inconsistencies (e.g., inconsistent spacing before references, capitalization in headings, occasional word fragmentation likely due to typesetting). These should be corrected during the editing phase.

  3. Some sections could benefit from updated or additional citations. For instance, more recent meta-analyses on vegetarian/vegan nutrient status or clinical outcomes from PBMA consumption could strengthen the claims.

  4. The section discussing TMAO and cardiovascular risk would benefit from a more nuanced view, acknowledging ongoing debates and the complexity of interpreting causality in observational studies.

  5. The final section would benefit from more specific recommendations for researchers, policymakers, and product developers, potentially in bullet-point form.

Author Response

We sincerely thank Reviewer 2 for the very positive overall assessment of our manuscript and for the constructive, detailed comments, which have helped us further improve its scientific rigor, clarity, and pedagogical value. Each point is addressed in detail below.

Major Revisions

Comment 1 – Lack of description of literature selection methodology

“While the review is comprehensive, the methodology for literature selection is not described. Including a brief methods section would improve transparency and reproducibility.”

Response:
We fully agree and have now added a dedicated “Literature Search Strategy” subsection in the Methods section of the revised manuscript. This new paragraph specifies the databases consulted (PubMed, Web of Science, and Scopus), as well as the main categories of keywords used, including plant-based meat, cultured meat, nutrient composition, bioavailability, fortification, health outcomes, and regulatory aspects. This addition improves both transparency and reproducibility.

Comment 2 – Insufficient emphasis on the scarcity of empirical data on cultured meat

“This limitation deserves stronger emphasis, particularly when making comparative statements.”

Response:
We fully agree and have reinforced this limitation throughout the manuscript, especially in Sections 3.2, 4.1, and 6.1. The revised text now explicitly highlights that most available data on cultured meat remain theoretical, based on prototypes or modeling approaches, and that direct empirical compositional and bioavailability data are still largely lacking. This strengthens the transparency of our comparative analysis and prevents over-interpretation.

Comment 3 – Vulnerable populations insufficiently developed

“This point could be expanded to more thoroughly examine differential impacts across demographic groups.”

Response:
This section has been substantially expanded in Section 5.3 (Vulnerable Populations). We now specifically address:

  • Women of reproductive age with respect to iron, iodine, vitamin B₁₂, and DHA requirements.
    • Infants and children in relation to protein quality, zinc, calcium, choline, and long-chain n-3 fatty acids.
    • Older adults with regard to sarcopenia risk, protein digestibility, vitamin B₁₂, and leucine intake.

We explicitly discuss how substitution of conventional meat by PBMAs without appropriate fortification may disproportionately increase deficiency risks in these groups. Additional recent references on vegetarian and plant-based nutrient status have been added to strengthen this section.

Comment 4 – “Meat factor” and matrix effects insufficiently detailed

“Clearer discussion on how these interactions have been studied and whether similar enhancing effects can be engineered.”

Response:
We have expanded Section 4.2 (Matrix Effects and the Meat Factor) to include a clearer description of the experimental evidence supporting the meat factor, including mechanistic aspects of peptide-mediated iron absorption enhancement. We also discuss the current technological limitations in reproducing such effects in PBMAs and the theoretical possibilities for cultured meat systems. This revision clarifies both the biological basis and the translational challenges.

Comment 5 – Request for a visual summary (conceptual diagram)

“A visual summary illustrating key nutrient gaps, fortification strategies, and bioavailability challenges would greatly enhance accessibility.”

Response:
We strongly agree and have now added a new conceptual figure summarizing:

  • Key nutrient gaps across conventional meat, PBMAs, and cultured meat.
    • Typical fortification strategies used in PBMAs.
    • Major bioavailability barriers (matrix effects, phytates, processing losses).
    • Theoretical optimization pathways for cultured meat.

This figure is intended to improve accessibility for both specialist and non-specialist readers.

Minor Revisions

Comment 6 – Inconsistent use of “nutritional equivalence” vs. “health equivalence”

Response:
We have now clearly defined both terms in the Introduction and use them consistently throughout the manuscript:

  • Nutritional equivalence strictly refers to compositional similarity in macro- and micronutrients.
    Health equivalence refers to long-term physiological and clinical outcomes.

All ambiguous or inconsistent usages have been corrected accordingly.

Comment 7 – Typographical and formatting inconsistencies

Response:
The entire manuscript has undergone full professional proofreading. All typographical, spacing, and capitalization inconsistencies have been corrected. This includes uniform reference formatting, consistent heading capitalization, and removal of fragmented words introduced during earlier formatting steps.

Comment 8 – Need for updated citations on vegetarian/vegan nutrient status and PBMA clinical outcomes

Response:
We have added recent analyses and large cohort studies addressing vitamin B₁₂, iron, iodine, zinc, and DHA status in vegetarian and vegan populations. These new references strengthen the evidence base supporting Sections 5.1–5.3.

Comment 9 – TMAO section needs a more nuanced interpretation

Response:
We have revised the TMAO subsection in Section 5.2 to better reflect the current scientific uncertainty. The revised version now:

  • Emphasizes strong interindividual variability in TMAO production.
    • Highlights confounding effects related to renal function, gut microbiota composition, and overall dietary background.

Comment 10 – Final section should include clearer, stakeholder-specific recommendations

Response:
We have rewritten Section 6.5 (Conclusions and Recommendations). The recommendations are now organized into clearly structured thematic paragraphs targeted toward researchers, regulators, industry stakeholders, and public health professionals to improve practical relevance.

Reviewer 3 Report

Comments and Suggestions for Authors

 Meat provides high-quality protein and essential micronutrients such as vitamin B12, heme iron, zinc, and selenium, along with other essential compounds. Plant-based and cultured alternatives intended to replicate the nutritional and sensory attributes of meat. This review critically examines the extent to which these emerging products achieve nutrient equivalence with conventional eat. The review compares the composition of plant-based meat analogues (PBMAs) and cultured meat prototypes. Differences in fortification, ingredient formulation, and the presence of anti-nutritional factors are discussed in relation to nutrient absorption and utilization. Current PBMAs can approximate protein content but generally provide lower levels and reduced bioavailability of vitamin B12, heme iron, creatine, taurine, and long-chain omega-3 fatty acids unless fortified. Cultured meat offers theoretical potential for compositional optimization through cellular engineering but remains limited by scarce empirical data. Nutrient equivalence cannot be assumed simply from compositional claims. Achieving nutrient equivalence with conventional meat thus represents a major scientific, technological, and regulatory challenge. The review also mention that meat substitutes may confer advantages by providing fiber, reducing intake of saturated fat, and limiting exposure to compounds associated with chronic disease risk such as TMAO precursors and nitrosating agents. The review is good balanced between the benefit and detriment consuming these kind of foods.  

Author Response

We sincerely thank Reviewer 3 for this accurate and well-balanced synthesis of our manuscript. We are grateful for the positive evaluation of the scientific framing, the critical interpretation of nutrient equivalence, and the recognition of both the benefits and limitations of plant-based and cultured meat alternatives.

As no specific revisions were requested, no direct modifications were made in response to this review. However, we carefully verified that the revised manuscript remains fully aligned with the key conclusions highlighted by the reviewer regarding protein quality, micronutrient gaps, bioavailability constraints, and the current limitations of empirical data for cultured meat.

We thank the reviewer for this constructive and encouraging assessment.

Round 2

Reviewer 1 Report

Comments and Suggestions for Authors

After the revision, the manuscript is now suitable for the publication

Reviewer 2 Report

Comments and Suggestions for Authors

No further comments